# Comparing the test performance of dried-blood-spot and plasma HIV recent infection testing samples in a nationally scaled sex worker programme in Zimbabwe

Brian D. Rice[1]*, Harriet S. Jones[2], Fortunate Machingura[3], Leah Gaihai[3], Memory Makamba[3], Thomas Chanyowedza[3], Panganai Masvikeni[3], Edward Matsikire[3], Primrose Matambanadzo[3], Sithembile Musemburi[3], Phillip N. Chida[3], Jeffery Dirawo[3], Owen Mugurungi[4], Sarah Bourdin[2], James R. Hargreaves[2], Gary Murphy[5], Frances M. Cowan[3,6]

1 University of Sheffield, School of Medicine and Population Health, Sheffield, United Kingdom, 2 Faculty of Public Health and Policy, London School of Hygiene and Tropical Medicine, London, United Kingdom, 3 Centre for Sexual Health and HIV/AIDS Research (CeSHHAR) Zimbabwe, Harare, Zimbabwe, 4 AIDS and TB Directorate, Ministry of Health and Child Care, Harare, Zimbabwe, 5 Independent Consultant, London, United Kingdom, 6 Faculty of Global Health, Liverpool School of Tropical Medicine, Liverpool, United Kingdom

* b.rice@sheffield.ac.uk

## Abstract

Recency testing can provide strategic insights as to whether a person newly diagnosed with HIV recently acquired their infection or not. To understand potential biases associated with HIV recency testing, we explored the extent sample type influences whether a person is assigned as being recent. Implementing a laboratory-based Recent Infection Testing Algorithm (RITA) across the Centre for Sexual Health and HIV/AIDS Research (CeSHHAR) key populations programme in Zimbabwe between October 2021 and January 2023, we compared plasma-based and dried-bloodspot (DBS) HIV recency samples. Over the study period, 24,976 individual female sex workers HIV tested, of whom 9.5% (2,363/24,979) newly tested HIV positive. Of these 2,363 women, 55.5% (1311/2,363) were offered and gave consent for a sample to be taken for DBS recency and viral load testing, among whom 11.7% (153/1,311) were classified as having a recent infection. A subset of 464 women were offered and consented to paired sample collection, among whom 10.1% (47) and 12.3% (57) of plasma and DBS samples, respectively, were classified as recent. Overall, categorical determination was good, with 97% of results concordant. Of 58 women with paired sample collection who had a test result classified as recent, 46 (79.3%) were concordant recent on both DBS and plasma, with 12 (20.7%) being discordant. Of these 12 women's samples, 11 were deemed long-standing by the plasma assay but recent by the paired DBS, and one deemed long-standing by DBS but recent by the paired plasma sample. On average, plasma samples had a higher normalised

**Data availability statement:** All relevant data for this study are publicly available from the Zenodo repository (https://doi.org/10.5281/zenodo.14984380).

**Funding:** This work was supported by The Bill and Melinda Gates Foundation (OPP119327 to BR, HSJ, FM, LG, MM, SB, JRH, GM, and FMC). The funders had no role in study design, data collection and analysis, decision to publish, or preparation of the manuscript.

**Competing interests:** The authors have declared that no competing interests exist.

optical density than DBS samples (mean difference of 0.53). Depending on use-case and setting, there are trade-offs when considering DBS or plasma-based samples between test performance and ease of implementation. Our data can help inform statistical adjustments to harmonise cut-offs on DBS and plasma assays, thereby improving the use and interpretation of recency assays in population-level HIV surveillance activities.

## Introduction

If we are to achieve sustainable HIV epidemic control, we need to better understand epidemic dynamics and improve the efficiency of programme service delivery. In gaining strategic insights of patterns and distributions of new HIV infections, and in prioritising prevention and testing interventions, there has been much discussion about the use of recency testing in programmatic setting [1–6]. Recency testing can help identify whether a person newly diagnosed with HIV recently acquired their infection (usually <one year) or not [1]. As a disproportionate number of HIV transmissions originate from people who have early HIV infection [7], identifying people or populations with recent infections has strategic potential to inform where interventions should be prioritised to reduce overall levels of transmission.

In relation to epidemic dynamics, it has been argued that there is a clear rationale for the use of recency assays for population-level HIV incidence estimation [2]. However, in relation to helping programmes focus limited resources to populations with the greatest potential benefit, it has been contended that their application for non-incidence use cases remains questionable [2].

In previous research we have shown the pragmatic application of laboratory-based HIV recency testing in programme settings in Kenya and Zimbabwe to be feasible [3,4]. Among calls for further research to better understand the potential biases associated with HIV recency testing [1,2,5,6], has been a request to explore the extent sample type influences whether a person is assigned as being recent [8]. In the context of a national scaled sex worker programme in Zimbabwe we explored whether conducting plasma-based HIV recency testing, in addition to our standard approach using DBS, impacted on recent HIV infection classification.

## Methods

### Ethics statement

Ethical approval was granted by the Medical Research Council of Zimbabwe (MRCZ/A/2244) and the London School of Hygiene & Tropical Medicine (14542 - 1). Informed consent was sought from all eligible women. All FSW testing newly HIV positive during the study period were eligible for enrolment in the study if aged 18 years and older and having received a confirmatory HIV positive test result, in line with Zimbabwe's national testing algorithm. Those under 18 years, with an indeterminate HIV test result or prior history of testing HIV-positive, or reported as being in receipt of anti-retroviral therapy, were excluded from enrolment.

## Study population and sample collection

We implemented a laboratory-based Recent Infection Testing Algorithm (RITA) across the Centre for Sexual Health and HIV/AIDS Research (CeSHHAR) key populations programme (formerly known as the Sisters with a Voice programme) in Zimbabwe between 01-10-2021 and 31-01-2023. The programme has been described elsewhere [9]. In summary, all women are assigned a unique identification number on first contact with the programme, with collected data being held centrally in an anonymized DHIS2 database. Since the programme's initiation in 2009, HIV testing coverage has increased substantially and those attending services are likely to be representative of female sex workers (FSW) in Zimbabwe [10–12].

In our study, a DBS sample was collected for each consenting woman. Samples were stored at clinic sites at room temperature in gas impermeable plastic bags with desiccant sachets to keep them dry. Samples were collected from sites every 14 days by a designated study driver and delivered to the Flow Cytometry Laboratory in Harare. In addition to a DBS sample, ethylenediaminetetraacetic Acid (EDTA) whole blood samples were also collected from all consenting women at Sisters clinics pre-identified as having the capacity to transport collected samples to the laboratory within 24 hours. Sites collecting plasma samples stored these in cool boxes with thermometers and trackers to monitor the temperature and timely delivery to the lab. Cooler boxes were collected at the end of each day by a delivery company. In addition to data collected routinely through the key population programme, anonymized enrolment details were collected and held on all participants in the central database, and were updated when RITA results were returned from the laboratory.

## Laboratory procedures

At the laboratory, EDTA whole blood samples were checked using quality control procedures and stored at 20 degrees Celsius. All available DBS and plasma samples were tested using the Maxim HIV-1 Limiting Antigen Avidity (LAg Avidity) Enzyme Immunoassay (EIA) (Maxim Biomedical, Bethesda, MD) in accordance with the product insert (DBS KIT CAT NO. 92003) [13]. Samples with a normalised LAg optical density (ODn) <2.0 underwent retesting in triplicate from a fresh dilution of the specimen to confirm the HIV LAg result as per the assay manufacturer's instructions for use. All DBS and plasma samples with a final ODn > 1.5 after retesting were classified as long-standing HIV infection.

Viral load testing was subsequently performed with a NucliSens ssay on all anti-HIV-1 positive DBS and plasma samples with a final ODn ≤ 1.5. All DBS and plasma samples with an ODn ≤ 1.5 and a viral load <1000 copies/ml were denoted long-standing infection, and those with an ODn ≤ 1.5 and a viral load ≥1000 copies/ml were classified as recent infection. Those samples with an ODn of <0.4 were retested with an Enzygnost Anti-HIV 1/2 Plus ELISA (Dade Behring, Inc., Marburg, Germany) test to confirm they were HIV-1 seropositive. All assays (including sample input volumes) were performed as per manufacturer's instructions for use.

## Data analysis

As informed by our RITA results, we first present HIV and recency tests results separately by DBS and plasma samples. A Bland-Altman plot was then applied to compare ODn values of plasma and DBS samples. STATA18 (Stata Corp, College Station, TX) was used for analyses.

## Results

Over the study period, 24,976 individual FSW HIV tested at CeSHHAR programme clinics, of whom 9.5% (2,363/24,979) newly tested HIV positive. Of these 2,363 women, 55.5% (1311/2,363) were offered and gave consent for a sample to be taken for DBS recency and viral load testing and were enrolled in the study. Among these women no samples were rejected, and 11.7% (153/1,311) were classified as having a recent infection.

Focusing on pre-identified clinics capable of plasma testing and transport, 464 women were offered and consented to paired sample collection (i.e., DBS and plasma, and viral load). Among these 464 women, plasma testing classified 10.1% (47) as having a recent infection, with DBS classifying 12.3% (57) (Table 1). Fifty-eight women had a paired sample collection with a test result classified as recent. Of these women, 46 (79.3%) were concordant recent on both DBS and plasma, 11 (19%) were recent on DBS only, and 1 (1.7%) was recent on plasma only (Table 1). Eleven samples were deemed long-standing by the plasma assay but recent by the paired DBS, and one deemed long-standing by DBS but recent by the paired plasma sample. Overall, there was 97.4% (452/464) concordance between the DBS and plasma samples (in addition to the 46 women concordant recent, 406 women were concordant long-standing).

Fig 1 compares ODn values of plasma and DBS samples. Of the 12 samples that were not in agreement for recent infection classification (pink dots), 3 had very different ODn values. The remaining 9 fell close to either side of the 1.5ODn cut-off. Fig 2 shows that, on average, plasma samples had higher ODn by a mean difference of 0.53 ODn (SD 0.98). Tests however did not show consistent variance across the range of ODn values, with ODn agreement being better at values below 2.

**Table 1. RITA classification of plasma and DBS samples.**

| | | DBS | | |
| --- | --- | --- | --- | --- |
| | | **Long-standing** | **Recent** | **Total (n & %)[1]** |
| **Plasma** | Long-standing | 406 | **11** | 417 (89.9%) |
| | Recent | **1** | 46 | 47 (10.1%) |
| | Total (n & %)[1] | 407 (87.7%) | 57 (12.3%) | 464 |

1 Presented as of total number of participants (464).

Discordant samples highlighted in bold.

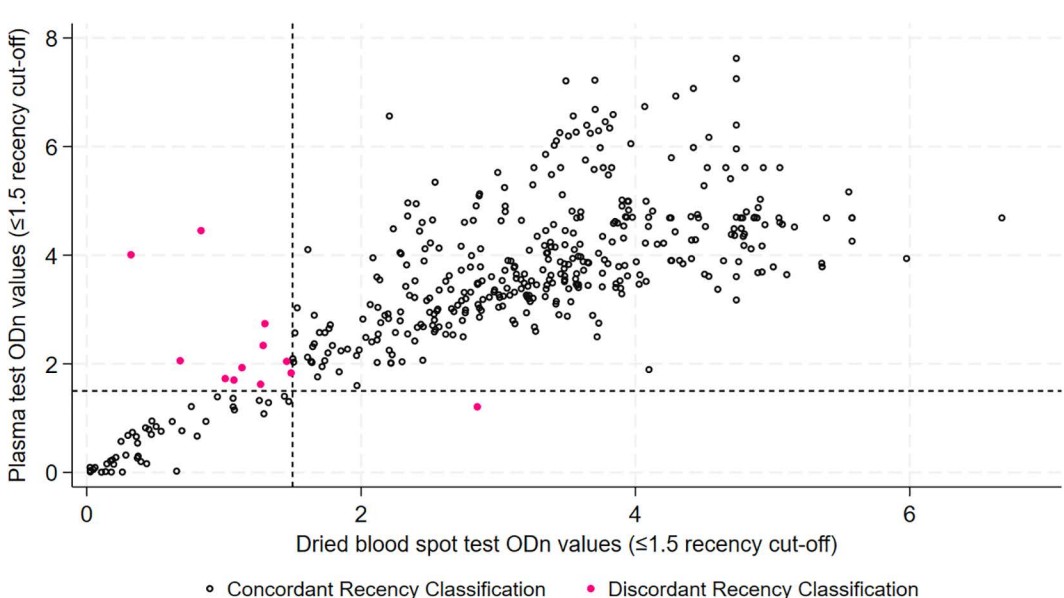

Fig 1. ODn values for plasma and DBS samples.

PLOS Global Public Health

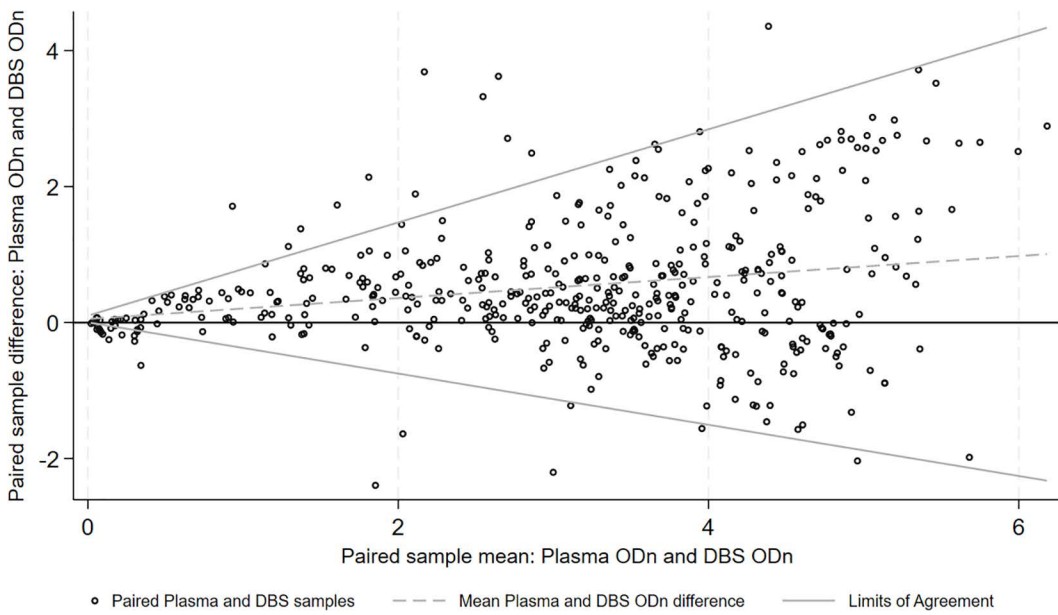

**Fig 2. Bland-Altman plot of plasma and DBS ODn on paired samples.**

## Discussion

To gain additional insights into routine HIV recent infection testing within a national scaled sex worker programme in Zimbabwe, we assessed agreement between DBS and plasma samples and explored implications for misclassification and implementation. Overall, categorical determination was good, with 97% of results concordant. However, we found the use of different sample types impacted on recent HIV infection classification. On average, plasma samples had a higher ODn than DBS sample (mean difference of 0.53), with ODn agreement being better at values below 2. This difference resulted in a higher percentage of women being classified as recently infected on their DBS sample versus plasma (12.2% versus 10.3%).

In a previous study test matching 100 plasma and DBS samples, ODn agreement was also shown to be greater at the lower range of values [8]. In this study, when the ODn was 2 the DBS value was 0.5 less than the plasma value [8]. Reporting DBS samples as being more likely to be classified as recent, the study's authors suggested DBS protocols could be adjusted to account for false recency results, or appropriate changes to assay cut-offs could be applied to account for higher rates of recency positive test results [8].

In relation to false recency results, we were unable to explore this in our study beyond including a viral load in our RITA. However, we can assume that both the plasma and DBS recent classified results are likely to include misclassification. We previously reported in Siaya county in Western Kenya that of eleven women classified as recent based on their LAg plasma samples and viral-load tests, one was reclassified as long-standing after a review of clinical documentation that showed she had initiated HIV treatment almost four years prior (recency misclassification of 9.1%) [3]. In Nairobi, Kenya, we previously reported that of 48 people testing recent based on their DBS sample and viral load one woman and one man were reclassified following antiretroviral therapy metabolite testing (misclassification of 4.2%) [3]. Through reducing false recency test results, the inclusion of metabolite testing in a RITA can strengthen the strategic potential of these data to help us understand transmission dynamics and where interventions should be prioritised. However, we have previously highlighted how the inclusion of metabolite testing is costly and makes the implementation of a RITA more challenging [4].

In relation to applying assay cut-offs, our data could be utilised to inform a statistical adjustment to harmonise cut-offs on DBS and plasma samples.

When conducting recency testing using DBS or plasma for different use-cases [2], there are factors in addition to test accuracy to be considered, for example implementation. In relation to three pilot studies of HIV recency testing we previously conducted in Kenya and Zimbabwe, we reported testing acceptance rates to differ by sample type (higher acceptance rates for DBS than plasma) [3,4]. We also reported how DBS sample collection was simpler than plasma sampling, as they could be transported and stored dry at room temperature (plasma requires maintenance of a cold chain during transportation and storage) [4]. In implementing the collection of both sample types, it was our experience in this study that DBS samples were easier to collect, transport and store than plasma samples. However, we also found DBS samples to be less timely than plasma due to the need for elution before testing. The issue of timeliness, however, was somewhat negated by the necessity for all recency tests to be analysed alongside viral load results for validation. This need for additional information, such as viral load, to interpret results, and to confirm HIV diagnosis, also equalises potential time gains from adopting recency point-of-care tests over laboratory-based testing for incidence estimation. A need to interpret recency point-of-care test results with caution was highlighted in a Ugandan study where archived plasma specimens with known periods of HIV seroconversion were used to validate the Asante HIV-1 rapid recency assay for use [14]. The authors of the study reported high test specificity (100% of long-standing infections were identified) but low test sensitivity (32% and 49% of recent infections were identified correctly in the two study laboratories) [14]. The authors suggest interlaboratory variability was due to differential interpretation of the test strip bands [14].

Laboratory-based recency testing, when conducted optimally, can distinguish recently acquired infection from long-standing infection among persons newly diagnosed with HIV. Depending on use-case and setting, however, there are trade-offs when considering DBS or plasma-based samples between test performance and ease of implementation.

When comparing DBS and plasma test results for the same assay, we need to consider potential differences in performance. We present information that will hopefully inform statistical adjustments to harmonise cut-offs on DBS and plasma assays, thereby improving the use and interpretation of recency assays in population-level HIV surveillance activities.

## Acknowledgments

We thank the women who enrolled in our study and those who visited CeSHHAR's Key Populations programme during the study period, all contributing data to this analysis. We thank everyone who has dedicated time to implementing CeSHHAR's Key Populations programme.

## Author contributions

**Conceptualization:** Brian D. Rice, Fortunate Machingura, Gary Murphy, Frances M Cowan.

**Data curation:** Harriet S Jones, Fortunate Machingura, Leah Gaihai, Memory Makamba, Thomas Chanyowedza, Jeffery Dirawo.

**Formal analysis:** Brian D. Rice, Harriet S Jones, Gary Murphy, Frances M Cowan.

**Funding acquisition:** Brian D. Rice, James R Hargreaves, Frances M Cowan.

**Investigation:** Harriet S Jones, Fortunate Machingura, James R Hargreaves.

**Methodology:** Brian D. Rice, Fortunate Machingura, Gary Murphy, Frances M Cowan.

**Project administration:** Fortunate Machingura, Leah Gaihai, Memory Makamba, Thomas Chanyowedza, Panganai Masvikeni, Edward Matsikire, Primrose Matambanadzo, Sithembile Musemburi, Phillip N Chida, Jeffery Dirawo.

**Resources:** Brian D. Rice, Owen Mugurungi, James R Hargreaves, Frances M Cowan.

**Supervision:** Brian D. Rice, Fortunate Machingura, Leah Gaihai, Memory Makamba, Primrose Matambanadzo, Owen Mugurungi, Sarah Bourdin, Frances M Cowan.

**Validation:** Brian D. Rice, Harriet S Jones, James R Hargreaves, Gary Murphy, Frances M Cowan.

**Visualization:** Harriet S Jones, James R Hargreaves, Gary Murphy, Frances M Cowan.

**Writing – original draft:** Brian D. Rice, Harriet S Jones, James R Hargreaves, Gary Murphy, Frances M Cowan.

**Writing – review & editing:** Brian D. Rice, Harriet S Jones.

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
