## [Decision Letter · Decision Letter 0]

15 Nov 2024

PGPH-D-24-02067

Comparing the test performance of dried-blood-spot and plasma HIV recent infection testing samples in a nationally scaled sex worker programme in Zimbabwe

Dear Dr. Rice,

Thank you for submitting your manuscript to PLOS Global Public Health. After careful consideration, we feel that it has merit but does not fully meet PLOS Global Public Health’s publication criteria as it currently stands. Therefore, we invite you to submit a revised version of the manuscript that addresses the points raised during the review process.

Please note that we have only been able to secure a single reviewer to assess your manuscript. We are issuing a decision on your manuscript at this point to prevent further delays in the evaluation of your manuscript. Please be aware that the editor who handles your revised manuscript might find it necessary to invite additional reviewers to assess this work once the revised manuscript is submitted. However, we will aim to proceed on the basis of this single review if possible. 

Could you please revise the manuscript to carefully address the concerns raised?

We look forward to receiving your revised manuscript.

Kind regards,

Helen Howard

Staff Editor

Journal Requirements:

2. Your current Financial Disclosure states, “OPP1120138”. However, your funding information on the submission form indicates that you received funding from “OPP119327”. Please indicate by return email the full and correct funding information for your study and confirm the order in which funding contributions should appear. Please be sure to indicate whether the funders played any role in the study design, data collection and analysis, decision to publish, or preparation of the manuscript.

3. In the online submission form, you indicated that "A de-identified dataset with variables included in this analysis can be made available on request to the Centre for Sexual Health & HIV/AIDS Research Zimbabwe, subject to ethical approval of a proposal.". 

3. Uploaded as supplementary information.

4. Please provide separate figure files in .tif or .eps format.

5. Please insert an Ethics Statement at the beginning of your Methods section, under a subheading 'Ethics Statement'. It must include:

1) The name(s) of the Institutional Review Board(s) or Ethics Committee(s)

2) The approval number(s), or a statement that approval was granted by the named board(s) 

3) (for human participants/donors) - A statement that formal consent was obtained (must state whether verbal/written) OR the reason consent was not obtained (e.g. anonymity). NOTE: If child participants, the statement must declare that formal consent was obtained from the parent/guardian.

Additional Editor Comments (if provided):

Reviewers' comments:

Reviewer's Responses to Questions

**Comments to the Author**

1. Does this manuscript meet PLOS Global Public Health’s publication criteria? Is the manuscript technically sound, and do the data support the conclusions? The manuscript must describe methodologically and ethically rigorous research with conclusions that are appropriately drawn based on the data presented.

Reviewer #1: Yes

2. Has the statistical analysis been performed appropriately and rigorously?

Reviewer #1: Yes

3. Have the authors made all data underlying the findings in their manuscript fully available (please refer to the Data Availability Statement at the start of the manuscript PDF file)?

Reviewer #1: Yes

4. Is the manuscript presented in an intelligible fashion and written in standard English?

Reviewer #1: Yes

5. Review Comments to the Author

Reviewer #1: Manuscript Number PGPH-D-24-02067

Reviewers Comments

Title: Comparing the test performance of dried-bloodspot and plasma HIV 2 recent infection testing samples in a nationally scaled sex worker programme 3 in Zimbabwe

Manuscript N0. PGPH-D-24-02067

Generally, the manuscript is well-written and organized

The study presents an important topic on recency testing and evaluates the use of DBS vs. Plasma for recency testing. The authors have described the study well in the introduction and its objective.

Line 65-66 (‘HIV 65 testing coverage has increased substantially and those attending services are likely to be representative of female sex workers (FSW) in Zimbabwe), the author’s statement highlighted in bold suggests that they are unsure of the population used in the study. Did you confirm that the enrolled participants were sex workers or were they persons seeking services from the mentioned clinics?

The authors have used the Viral load test, HIV Lag, Avidity and ELISA test in determining the thresholds for either recent or long-term infection. However, the authors did not mention which ELISA kit was used, or the manufacturer’s name, similar to the HIV LAg Avidity test used in this study.

The authors have not mentioned the sample input volume used in the testing for HIV Viral load, ELISA, and HIV LAg Avidity.

Could it be that the threshold used for HIV viral load testing using the DBS matrix contributed to the over classification of more recent infections compared to plasma Viral load?

Suggestion for your consideration:

See line 105-Can the authors consider presenting the data on the 58 recent infections in a 2by-2 table and have the sensitivity and specificity determined for each sample matrix?

Could the results have been different if DBS were accorded similar storage treatment as plasma from collection to storage instead of having them stored at room temperature for 14 days? Could this have been a limitation in the design of this study?

Recommendation: Areas that need review by the authors are considered minor.

6. PLOS authors have the option to publish the peer review history of their article (what does this mean?). If published, this will include your full peer review and any attached files.

**Do you want your identity to be public for this peer review?** For information about this choice, including consent withdrawal, please see our Privacy Policy.

Reviewer #1: No

---

## [Decision Letter · Decision Letter 1]

18 Jun 2025

PGPH-D-24-02067R1

Comparing the test performance of dried-blood-spot and plasma HIV recent infection testing samples in a nationally scaled sex worker programme in Zimbabwe

Dear Dr. Rice,

Thank you for submitting your manuscript to PLOS Global Public Health. After careful consideration, we feel that it has merit but does not fully meet PLOS Global Public Health’s publication criteria as it currently stands. Therefore, we invite you to submit a revised version of the manuscript that addresses the points raised during the review process.

Details of sample processing and which sample used for which test are necessary. Sub headings might help

Line 87: This statement starting with “Samples with a normalised optical density (ODn) <2.0…” seems very abrupt. A subheading or context may be necessary

Line 107: “the authors state “ of the 58 women with paired samples…” Where is this number coming from? Plasma identified 47 and DBS identified 57.

Little or no statistical analyses details

We look forward to receiving your revised manuscript.

Kind regards,

Muki S. Shey, Ph.D

Academic Editor

Additional Editor Comments (if provided):

Reviewers' comments:

Reviewer's Responses to Questions

**Comments to the Author**

1. If the authors have adequately addressed your comments raised in a previous round of review and you feel that this manuscript is now acceptable for publication, you may indicate that here to bypass the “Comments to the Author” section, enter your conflict of interest statement in the “Confidential to Editor” section, and submit your "Accept" recommendation.

Reviewer #2: All comments have been addressed

Reviewer #3: (No Response)

2. Does this manuscript meet PLOS Global Public Health’s publication criteria? Is the manuscript technically sound, and do the data support the conclusions? The manuscript must describe methodologically and ethically rigorous research with conclusions that are appropriately drawn based on the data presented.

Reviewer #2: Yes

Reviewer #3: Partly

3. Has the statistical analysis been performed appropriately and rigorously?

Reviewer #2: Yes

Reviewer #3: N/A

4. Have the authors made all data underlying the findings in their manuscript fully available (please refer to the Data Availability Statement at the start of the manuscript PDF file)?

Reviewer #2: Yes

Reviewer #3: Yes

5. Is the manuscript presented in an intelligible fashion and written in standard English?

Reviewer #2: Yes

Reviewer #3: Yes

6. Review Comments to the Author

Reviewer #2: while the findings are well explained, their implications could be better developed—particularly how the ODn shift might influence programmatic thresholds or the statistical adjustment of incidence estimates.

2.2. Methodological Clarity

The methods are mostly clear, but a few issues need elaboration:

• Paired Sampling Selection Criteria: What were the criteria for selecting women for paired sampling? Were there systematic differences between the full cohort and the 464 paired-sample participants?

• Viral Load Testing Consistency: The RITA requires viral load data. Clarify whether viral load was always measured from plasma or whether DBS VL was used in some cases. Variability here could affect classification outcomes.

• ODn Threshold: Was the same ODn cut-off (1.5) applied to both DBS and plasma without adjustment? The manuscript implies this but should state it explicitly.

• Handling of Missing or Invalid Results: No samples were reportedly rejected—this is uncommon. Please confirm how sample integrity and validity were assessed and whether any samples were excluded from analysis due to assay failure.

2.3. Interpretation of Results

The authors present an honest and cautious interpretation. Still, a few aspects could be strengthened:

• Quantitative Summary of Discrepancy: Expand on the degree and direction of discordance. For instance, 11 of 12 discordant cases were recent by DBS and long-standing by plasma—does this suggest DBS is systematically more sensitive, or perhaps more prone to false recency?

• Figure Interpretation: Figure 1 and 2 are informative but would benefit from:

o Labeled axes with units and cut-off lines clearly indicated.

o A short caption describing key takeaways from the figure (e.g., “pink dots” are discordant).

o Clearer legends differentiating recent/long-standing classifications.

• Bland-Altman Plot: Consider adding limits of agreement and confidence intervals to better convey the precision of the ODn difference.

2.4. Literature Context

The discussion references appropriate prior studies, including the known higher likelihood of DBS to yield “recent” classifications. However, it would be helpful to:

• Include a summary of previous quantitative comparisons of ODn values between DBS and plasma (e.g., mean differences, variability).

• Discuss whether the observed mean ODn difference (0.53) aligns with previous validation studies and manufacturer documentation.

• Cite examples of statistical methods used in the literature to adjust for these differences (e.g., regression calibration, cut-off optimization).

3. Minor Comments

• Abstract: Consider stating the key numerical result from the ODn comparison (mean difference of 0.53) directly in the abstract.

• Ethics: Clarify how confidentiality and anonymity were ensured given the sensitivity of the population (e.g., storage of consent forms, coded ID systems).

• Terminology: Replace "dis-concordant" with the standard "discordant" throughout.

• Typographical Issues: Remove repeated sentences in the Methods section (lines 63–69 and 75–80 appear nearly duplicated).

• Table 1: Add row and column percent totals for better interpretability.

• Footnotes: Ensure all figures and tables have explanatory footnotes or legends.

5. Discussion

5. 1. Clarity of Objectives

The manuscript starts by mentioning the comparison of DBS and plasma for recency testing, but the rationale for why this comparison is necessary or important is not fully articulated upfront. Clearly state the objectives at the beginning of the section. For example, "This study aimed to assess the agreement between DBS and plasma samples for recent HIV infection testing and explore implications for misclassification and implementation."

5.2. Results Interpretation

The findings regarding concordance (97%) and differences in ODn values are clearly stated, but the implications of these findings on programmatic decisions or surveillance estimates are not fully explored. Elaborate on how a 2% discordance or the difference in ODn values could influence incidence estimation, public health decision-making, or resource allocation.

5.3. Discussion of Misclassification

The manuscript references misclassification but doesn’t quantify it for the current study or differentiate clearly between false recency due to sample type vs. treatment history. Provide an estimate of potential misclassification in your own dataset or clarify why it could not be estimated (e.g., lack of ART metabolite testing). Highlight this limitation explicitly.

Minor Comments

• Line 134–135: Consider rewording "a higher percentage of women being classified as recent positive" to "a higher percentage of women were classified as recently infected."

• Line 141–144: "We were unable to explore this..." would benefit from clearer justification. Could mention logistical or ethical constraints.

• Line 151: "Preferably in combination with other complementary data" is vague. What type of data is meant here?

• Lines 157–160: Good point about storage, but could be stronger if supported with cost or logistical data.

• Line 167: Consider citing specific limitations of point-of-care tests mentioned in references [14,15].

Reviewer #3: Comments to Author:

Title: Comparing the test performance of dried-blood-spot and plasma HIV recent infection testing samples in a nationally scaled sex worker programme in Zimbabwe

Overview and general recommendation:

In this short report manuscript, Rice BD and collaborators compared both dried-blood-spot and plasma for detecting recent HIV infection at a large scale sex worker programme in Zimbabwe. They found a concordance of 97% which is very good. They proposed that the used of one of these samples depend on the use-case and setting. Recency test in HIV infection is critical. Most infection involve people who recently acquired the virus as the amount of the virus in their body is extremely high. Therefore identify such individuals is crucial in the fight against the infection; therefore the current study carries some good interests.

Major Comment:

My major comment turns around the validation of the test used. Usually, when comparing two methods or two samples, there should be a gold standard, something to validate the results. The authors should not rely on the test performances provided by the manufacturer, but should have made a prior validation of the test used and the best example would be to use sample from individuals with known time of infection. While 47 women were found to be recently infected using plasma, there were 57, when the authors used whole DBS. The difference is high and no one can tell for sure if these 10 individuals were recently infected or not. It may turn out that this difference is partly the consequence of the overall sensitivity and/or specificity of the test used.

In addition, results are a bit confusing; how was the 97% agreement obtained? Also, the authors mentioned 58 women with paired samples who had result classified as recent….where this number of 58 from? I tough there were 464 individuals who provided the two biology specimen ….. and again the concordance in this 58 women is different from the initial 97% concordance. The authors should please clarify the results.

7. PLOS authors have the option to publish the peer review history of their article (what does this mean?). If published, this will include your full peer review and any attached files.

**Do you want your identity to be public for this peer review?** For information about this choice, including consent withdrawal, please see our Privacy Policy.

Reviewer #2: No

Reviewer #3: **Yes: **Tongo Marcel

---

## [Editor Report · Decision Letter 2]

7 Aug 2025

PGPH-D-24-02067R2

Comparing the test performance of dried-blood-spot and plasma HIV recent infection testing samples in a nationally scaled sex worker programme in Zimbabwe

Dear Dr.Rice

Thank you for submitting your manuscript to PLOS Global Public Health. After careful consideration, we feel that it has merit but does not fully meet PLOS Global Public Health’s publication criteria as it currently stands. Therefore, we invite you to submit a revised version of the manuscript that addresses the points raised during the review process.

Please address comments from Reviewer 2 and Reviewer 3 in the previous decision of the manuscript.

We look forward to receiving your revised manuscript.

Kind regards,

Muki S. Shey, PhD

Academic Editor

Journal Requirements:

Additional Editor Comments:

Please address comments from Reviewer 2 and Reviewer 3 in the previous decision of the manuscript.
---

## [Decision Letter · Decision Letter 3]

15 Oct 2025

Comparing the test performance of dried-blood-spot and plasma HIV recent infection testing samples in a nationally scaled sex worker programme in Zimbabwe

PGPH-D-24-02067R3

Dear Dr Rice,

We are pleased to inform you that your manuscript 'Comparing the test performance of dried-blood-spot and plasma HIV recent infection testing samples in a nationally scaled sex worker programme in Zimbabwe' has been provisionally accepted for publication in PLOS Global Public Health.

Best regards,

Muki S. Shey, PhD

Academic Editor

Reviewer Comments (if any, and for reference):

Reviewer's Responses to Questions

**Comments to the Author**

1. If the authors have adequately addressed your comments raised in a previous round of review and you feel that this manuscript is now acceptable for publication, you may indicate that here to bypass the “Comments to the Author” section, enter your conflict of interest statement in the “Confidential to Editor” section, and submit your "Accept" recommendation.

Reviewer #2: All comments have been addressed

Reviewer #3: All comments have been addressed

2. Does this manuscript meet PLOS Global Public Health’s publication criteria? Is the manuscript technically sound, and do the data support the conclusions? The manuscript must describe methodologically and ethically rigorous research with conclusions that are appropriately drawn based on the data presented.

Reviewer #2: Yes

Reviewer #3: Yes

3. Has the statistical analysis been performed appropriately and rigorously?

Reviewer #2: Yes

Reviewer #3: N/A

4. Have the authors made all data underlying the findings in their manuscript fully available (please refer to the Data Availability Statement at the start of the manuscript PDF file)?

Reviewer #2: Yes

Reviewer #3: Yes

5. Is the manuscript presented in an intelligible fashion and written in standard English?

Reviewer #2: Yes

Reviewer #3: Yes

6. Review Comments to the Author

Reviewer #2: The author has satisfactorily addressed all reviewer comments and incorporated the necessary revisions. The responses are comprehensive, and the manuscript has improved significantly in clarity, rigor, and presentation. I therefore recommend that the manuscript be accepted for publication in its current form.

Reviewer #3: I am satisfied with the comments provided y the authors

7. PLOS authors have the option to publish the peer review history of their article (what does this mean?). If published, this will include your full peer review and any attached files.

**Do you want your identity to be public for this peer review?** For information about this choice, including consent withdrawal, please see our Privacy Policy.

Reviewer #2: **Yes: **sinaye ngcapu

Reviewer #3: **Yes: **Tongo Marcel
